# Autoencoder Composite Scoring to Evaluate Prosthetic Performance in Individuals with Lower Limb Amputation

**DOI:** 10.3390/bioengineering9100572

**Published:** 2022-10-18

**Authors:** Thasina Tabashum, Ting Xiao, Chandrasekaran Jayaraman, Chaithanya K. Mummidisetty, Arun Jayaraman, Mark V. Albert

**Affiliations:** 1Department of Computer Science and Engineering, University of North Texas, Denton, TX 76203, USA; 2Department of Information Science, University of North Texas, Denton, TX 76203, USA; 3Max Näder Lab for Rehabilitation Technologies and Outcomes Research, Shirley Ryan AbilityLab, Chicago, IL 60611, USA; 4Department of Physical Medicine and Rehabilitation, Northwestern University Feinberg School of Medicine, Chicago, IL 60611, USA; 5Department of Biomedical Engineering, University of North Texas, Denton, TX 76203, USA

**Keywords:** autoencoder, principal component analysis, lower limb amputation

## Abstract

We created an overall assessment metric using a deep learning autoencoder to directly compare clinical outcomes in a comparison of lower limb amputees using two different prosthetic devices—a mechanical knee and a microprocessor-controlled knee. Eight clinical outcomes were distilled into a single metric using a seven-layer deep autoencoder, with the developed metric compared to similar results from principal component analysis (PCA). The proposed methods were used on data collected from ten participants with a dysvascular transfemoral amputation recruited for a prosthetics research study. This single summary metric permitted a cross-validated reconstruction of all eight scores, accounting for 83.29% of the variance. The derived score is also linked to the overall functional ability in this limited trial population, as improvements in each base clinical score led to increases in this developed metric. There was a highly significant increase in this autoencoder-based metric when the subjects used the microprocessor-controlled knee (*p* < 0.001, repeated measures ANOVA). A traditional PCA metric led to a similar interpretation but captured only 67.3% of the variance. The autoencoder composite score represents a single-valued, succinct summary that can be useful for the holistic assessment of highly variable, individual scores in limited clinical datasets.

## 1. Introduction

Therapeutic interventions to improve quality of life are often measured indirectly through a series of clinical scores that are well-defined and understood among clinicians, but present challenges. The goal of most interventions is to improve overall health, and the array of measures often available to assess an individual does not reflect this unified goal. Additionally, when evaluating individuals, single clinical measures of therapeutic response may be highly variable, but a significant, overall improvement in mobility can often be readily observed from all the obtained measures. Creating summary metrics can provide a more stable and unified assessment of individual performance; this is particularly valuable for limited clinical population studies, for example, in studies involving extensive data collection for lower limb amputees.

Lower limb amputation dramatically changes an individual’s mobility and interferes with many more aspects of daily living. Due to an individual’s inability to engage in many activities, their quality of life is severely affected [1]. Measuring outcomes of mobility and functionality are important, but often the underlying goal is to impact an overall sense of health; improving the overall quality of life (QoL) is the main guiding objective of rehabilitation [2,3]. However, quality of life is complex to measure as there is no well-established sole definition of it [4]. The multidimensionality and subjectivity of QoL make it more difficult to interpret. Even though there is no consensus on a single definition of QoL, a substantial contribution can be made if strategies are found to present clear and comparable indications in individuals of the overall health and well-being of people.

Directly measured and summarized clinical scores are critical for evaluating health outcomes using traditional statistical techniques as well as more complex predictive models [5]. For instance, Wang et al. predicted the gait normalcy index from nine gait variables extracted through inertial measurement units. The authors showed that their proposed method can quantify patients’ gait progress by comparing them with healthy subjects [6]. Multilayer neural networks in deep learning are capable of taking into account the complex nonlinear mathematical relationships among measures that challenged previous outcome modeling strategies [7]. In the biomedical field, deep learning methods increasingly perform a vital role in inference as part of smart healthcare systems [8]. Yang et al. proposed a convolutional neural network and predicted an expanded disability status scale (EDSS) to measure the severity of multiple sclerosis [9]. In particular, as an unsupervised machine learning method, deep autoencoders use deep neural networks to encode data into a reduced representation and then evaluate the quality of that encoding by reconstructing the original data [10]. Such reduced encodings provide a valuable, succinct summary of the often highly correlated and noisy original dataset. Much like traditional factor analysis, such reduced representations can be leveraged for better learning or interpretation of complex datasets. Autoencoders have demonstrated good performance in various applications, including the assessment of VR sickness [11], non-intrusive speech quality [12], and lung cancer [13]. The dimensionality reduction ability of autoencoders can be utilized to have clear 2D or 3D visualizations of data samples, helping to visualize relationships among samples represented by high-dimensional vectors [14].

A number of factor analysis strategies can be applied to reduce the dimensionality of a problem to find latent variables of interest or representations to improve prediction. One of the most common strategies, principal component analysis (PCA), has been used to accomplish this reduction by capturing underlying factors within the data that account for the correlations between variables [15,16]. PCA-based factor analysis techniques and deep autoencoders can be used to identify latent factors in highly correlated datasets [17,18]. However, PCA is limited as a linear transformation technique; additionally, PCA relies on strong mathematical assumptions, such as forcing orthogonality between projections. This leads to factors later in the rank that are challenging to interpret and that do not succinctly capture the statistical relationships present in the data [19]. Moreover, real-life datasets have nonlinear complex relationships, where linear methods are incapable of capturing the association between the variables. For instance, Krakovska et al. compared linear with nonlinear methods in the feature selection process and showed that a nonlinear approach is more consistent [20]. Previous studies have demonstrated that the existence of nonlinear association in health science is high [21,22]. As a consequence, linear methods might fail to capture the relationships of highly varied data.

However, autoencoders are capable of learning linear and nonlinear relationships of variables in unsupervised settings [23,24]. Recent studies have quantitatively compared autoencoders with PCA and showed that autoencoders perform better in capturing latent representations of data [25,26]. In searching for a single metric representing a person’s health among a series of health indicators, autoencoders are likely to outperform PCA for summarizing complex, nonlinear relationships, as often occur in health assessment measures.

Addressing the overall health and ability of individuals with lower limb amputations is particularly important. The number of lower limb amputations is increasing significantly, and it will more than double by 2050 to 3.6 million [27]. According to the National Limb Loss Resource Center, this growth is due to the aging of the population and vascular diseases, including diabetes and peripheral arterial disease [28]. Every year in the US, new amputations from dysvascular disease are rising; approximately 185,000 undergo limb amputations [29], and a majority of the lower limb amputations are caused by diabetes [30,31], with a 55% chance of requiring repeated amputation within 2–3 years [32]. Therefore, the direct evaluation of individuals is crucial. To assess individuals’ gait parameters, Li et al. proposed a foot pose estimation method using a pair of wearable shoes that capture kinetic and kinematic measurements [33]. Nonlinear techniques of machine learning and deep learning have shown promising results in detecting abnormal gait patterns [34,35,36]. However, the main contribution of this study is that we propose an approach to creating a single score that directly measures the two prosthetics of each individual. The scores show statistically significant improvements when the subjects used a microprocessor knee.

The primary goal of this study was to evaluate the feasibility of using a composite score for interpreting data from prosthetics research for evaluating overall performance. We demonstrate the use of a metric to assess overall performance in this population using a PCA and an autoencoder. As a result, we created a single composite metric to measure improvement in patient performance in relation to their individual scores. To validate the test of this approach, we utilized data from a microprocessor-controlled knee (MCK)–mechanical knee (MK)/mechanical knee (MK)–microprocessor-controlled knee (MCK) experimental design. The outcome of the autoencoder single composite score is consistent with previously published research [37]. Jayaraman et al. showed that the gait quality of participants statistically improved using the microprocessor-controlled knee. In studies with many metrics and small sample sizes, a single composite metric might help clear the directionality of the interventions. We propose a seven-layer autoencoder, and the hidden dimension is one. The autoencoder takes eight different clinical test scores and reduces them into one score. We show that the autoencoder approach can preserve maximum information that is more than 83.29% in the test set, and with no supervision, it is capable of deriving the hidden representation of each subject’s conditions.

## 2. Methods

### 2.1. Study Design

This study is a secondary analysis of data collected from an outcome research study [37] involving an MCK and MK at pre- and post-intervention. The study involved randomization with a repeated measures/crossover design to compare devices (MCK and MK) on everyday function, social interaction, and quality of life in subjects with transfemoral amputation. Subjects were randomly assigned to one of the two arms of the study: the MCK or the MK. Assessments for balance and mobility were gathered at the time of enrollment and at the end of the trial. Taking these scores, we designed a 7-layer-deep neural network and compared the overall performance achieved for each subject between the MK and the MCK.

### 2.2. Participants

A total of 10 subjects with transfemoral amputations were recruited for the previous clinical study from the inpatient and outpatient amputee clinics of the Shirley Ryan AbilityLab and Northwestern Memorial Hospital (Chicago, IL, USA). Nine out of the 10 enrolled subjects completed the study; one subject chose to withdraw due to personal reasons. The mean age of the subjects was 63 years and the mean amputation duration was 5.8 years. All subjects provided informed consent prior to participation in the study, which was approved by the Northwestern University Institutional Review Board. All study procedures were carried out following the standards listed in the Declaration of Helsinki, 1964. No adverse events were reported during the entire study duration.

### 2.3. Data and Statistical Analysis

Assessments at the beginning and end of each home trial included performance measures and self-reported measures to assess function, mobility, balance, and quality of life. Performance measures consisted of a 10-m walk test (10 MWT), 6-min walk test (6 MWT), the BERG balance test (BERG), the Four Square Step Test (FSST), and the Timed UP and Go test (TUG). Subjects could use a cane/walker if needed during these tests. Self-reported measures included the following paper-based questionnaires: (i) Amputee Mobility Predictor (AMP), (ii) Modified Falls Efficacy Scale (MFES), and (iii) the Prosthesis Evaluation Questionnaire (PEQ). The questionnaires were scored as per the standard scoring guides associated with each survey. The differences between the two conditions were evaluated using a fit generalized linear model, with indications of statistical significance using a *p*-value threshold of 0.05.

Both PCA and the described autoencoder were used to aggregate the results of the clinical scores into a single metric to test the overall impact of the choice of prosthetic leg. PCA is commonly used and deep autoencoders are increasingly being used to reduce the number of factors needed to represent higher-dimensional data [38,39,40]. In this case, the 8 assessed metrics were reduced in dimensionality to only 1 factor so that the differences between the subjects before and after the introduction of the MCK could be compared along one summary dimension, with less influence of noise that would be present in a single measure.

### 2.4. Autoencoder Architecture

An autoencoder is an unsupervised neural network that can represent data in new reduced dimensions by learning the linear and nonlinear relationships of the data [41,42]. The objective of an autoencoder is to learn a lower dimensional representation of the data where the output is a reconstruction of the input. An autoencoder has three parts: the encoder, the code, and the decoder. The encoder part learns the new feature space *z* of the input *x*, the code is the new feature space, and the decoder part tries to decode the new feature space *z* into the original space *x*′. The goal is to learn the new feature space while losing the least amount of information. To evaluate the model in this study, we used the explained variance. The explained variance measures the amount of information the model captures.
*z* = *h*_1_ (*w*_1_*x* + *b*_1_) (1)
*x*′ = *h*_2_ (*w*_2_*z* + *b*_2_) (2)

In Equations (1) and (2), *h* represents the activation functions, *w* represents the weights, and *b* represents bias. Figure 1 shows the architecture of the autoencoder.

We created an encoder and a decoder, with both of them consisting of two fully connected layers, including a dropout layer. The input is the 8 test scores. The encoder contains three layers composed of two fully connected layers with 8 and 24 neurons and a dropout layer with a 0.4 rate. Similarly, the decoder includes a dropout layer with a 0.4 rate and two fully connected layers with 24 and 8 neurons, consecutively. The composite score is the output of the single central neuron. Notably, it is unusual for autoencoder architectures to have only one neuron for the encoding; however, given the goal of this analysis and the limited data, we used only a single neuron. In this way, the network was able to compress the eight scores to one, and again reconstruct the data back to the original 8 scores with as high fidelity as possible. The activation functions for the layers with 24 neurons were relu, and the code layer was linear. The autoencoder structure is tabulated in Table 1. The loss function utilized was the mean squared error, as shown in Equation (3).
(3)$$L=\frac{1}{N}\sum (x-x^{\prime }{)}^{2}\;$$

The learning rate was set to 0.01 and the Adam optimizer was used. All the features were normalized by standard scaling. The model was evaluated using the variance as defined in Equation (4).
(4)$$s^{2}=\frac{1}{\left(n-1\right)}\sum (x-x^{\prime }{)}^{2}\;$$

All the hyper-parameters were fixed using a grid search. We searched the activation functions relu, tanh, sigmoid, and linear. For the number of neurons in the second layer, we searched 10, 12, 16, 24, and 32. While searching the hyper-parameters, we set the same number of neurons and activation functions for the decoder. Since the dataset was small, a deeper network may have resulted in an overfitted model, and we only tried one fully connected layer after the input.

## 3. Results

Comprehensive assessments were performed both for quantitative and qualitative measures. In Section 3.1, the clinical performance and individual self-reported metric results are briefly illustrated. Further, Section 3.2 and Section 3.3 demonstrate the composite scores of the proposed deep autoencoder and PCA.

### 3.1. Performance and Self-Reported Measures

The gait speeds based on the 10-m walk tests (10 MWT) were significantly higher for the MCK compared to the values for the MK. The group average with the standard deviation in 10 MWT was 15.36 (±7.2) seconds while using the MCK in comparison to the MK, with 22.27 (±9.72) seconds. However, there were no significant differences in the distances during the 6-min walk tests (6 MWT), Four Square Step Test (FSST), and TUG between the conditions (*p* > 0.05). The average BERG scores for the MCK was 44 m, while the average was 37 m for the MK. The group average of the Ambulation PEQ score was 81.92 (±18.74) for the MCK, whereas it was 60.63 (±18.75) for the MK. The subject-reported scores for both the Amputee Mobility Predictor (AMP) survey and the Modified Falls Efficacy Questionnaire (MFES) were higher on average for the MCK compared to the MK. The Prosthetics Evaluation Questionnaire (PEQ) covered modules on ambulation, appearance, frustration, perceived response, residual limb health, social burden, sounds, utility, and well-being. These self-reported measures (PEQ-A, MFES) indicate that the subjects were able to walk on different surfaces and had a considerably better-reduced fall risk with the MCK condition.

### 3.2. Autoencoder Composite Score

The model was evaluated by two runs of 90% data as the training set and 10% as the test set. Each run test set was completely different. The mean variance was 83.29%, and the mean reconstruction error was 0.18 for the test set. Given the varying levels of significance in the individual assessment scores, we used an autoencoder to assess the overall impact of using the MCK. To visualize the results, the dimensionality of the eight assessment scores for each subject was reduced to only one dimension per subject. Figure 2 represents the transformed scores for each subject, one after using the MK and one for the CK. Each subject’s composite score difference is represented by a straight arrow from the MK to the MCK. Due to the fact that the score increased with any improvement in the eight metrics, the composite score represents overall health with regard to these measures. The average and standard error composite scores for the MK and CK along this dimension were 0.29 (±0.89) and −1.92 (±1.13), respectively, representing greater overall health under the CK condition. When performing repeated measures ANOVA for these two conditions, the change in the composite score was highly significant (*p* < 0.001) which, as expected, reflects the aggregate contribution of a number of marginally significant scores. Table 2 illustrates the means and standard deviations of all eight test results and one summary score of the autoencoder (AE) for each condition.

### 3.3. PCA Composite Score

PCA is also an approach that represents data in fewer dimensions, and we applied this reduction for comparison to the autoencoder approach. The dimensionality of the eight assessment scores for each subject was reduced to only two dimensions per subject, using PCA for visualization below. Figure 3 represents the spread of these transformed scores with two points for each subject, one after using the MK and one for the MCK. To understand what these two summary dimensions represent, the load factors for the transformations that constitute the PCA axes are shown in Table 3. The X-axis represents the first component, which accounted for 67.3% of the variance in the clinical scores; thus, we focus on this factor for interpretation. Notably, in the PCA, whether we extracted two dimensions or only one, the dimensions remained the same—this was not the case with the autoencoders. The first principal component also represents overall health, since the load factors are positive for measures that should increase with improved health and negative for those that should decrease with improved health (Table 1). The average and standard error composite scores for the MK and MCK along this dimension were −0.702 (±0.634) and 0.873 (±0.738), respectively. According to the ANOVA of the values along this axis, the change in the composite score was highly significant (*p* < 0.001). The Y-axis shows the second PCA component, which only represents 9.8% of the variance in the measures (only one-seventh as informative as the X-axis component) and is less amenable to interpretation as it is orthogonal to the first component. Thus, in Figure 3, we compressed the Y-axis to emphasize the role of the first component. The CK scores are shown farther along the higher-performance direction of the X-axis.

## 4. Discussion

The primary goal of this study was to test the feasibility of a deep neural network autoencoder to evaluate prosthetic performance. This was applied in this case to evaluate the potential of prosthetic devices using clinical outcomes and self-reported measures in a small sample of dysvascular transfemoral amputees. In order to provide a well-grounded overall assessment of improvement for subjects, these eight measures were combined into a single composite score, which showed a highly significant change in health with respect to these measures in the direction of improvement, highlighted by changes for all nine individual subjects along this metric with a *p* < 0.001. A consistent change in the direction of improvement along each of these scores is much clearer in this case, which is particularly important with limited, potentially noisy clinical observations.

We find the composite score results of the proposed autoencoder particularly compelling and suggest it as a standard approach for summarizing overall changes in subject performance relative to a large set of assessment scores. This is particularly relevant for studies with relatively few subjects and a large number of metrics, as individual tests of improvement in assessments may lack sufficient data to make a clear statement of statistical significance. An autoencoder is a standard method of summarizing high-dimensional data with fewer dimensions and is well-known in data compression, factor analysis, and data visualization, with data analysis tools readily available to apply to clinical data. Importantly, an autoencoder does not use knowledge of the device group (MK or MCK) in determining the axes to summarize the scores, thus any differences between groups are clearly an effect of the conditions rather than the technique itself. The autoencoder provides a simple way of distilling a series of scores into a single unified metric to assess changes.

Additionally, we compared the results with PCA. The first PCA dimension accounted for over 67% of the variation in the scores between subjects, whereas 83% of the variation in the scores was attributed to the proposed autoencoder. The results demonstrate the superior performance of the autoencoder in comparison to PCA, which was expected, as autoencoders can capture nonlinear relationships in data. Hinton et al. showed that deep autoencoders are better for data compression than shallower approaches for complex statistical relationships [39]. Moreover, autoencoders have flexibility, which can be structured as needed for any specific kind of problem. In this case of limited data, a simple autoencoder for the eight inputs for each subject and condition, distilled to a single neural representation, and multiple layers to capture the nonlinear relationships led to the model choices made in this analysis.

The one summary dimension accounted for over 83.29% of the variation in the scores between subjects, providing a compelling metric matching the intuition of the overall improvements in mobility that can be seen even with small samples. The composite score led to a clearer statistical interpretation of the overall improvement in mobility that was consistent for each subject in the study. The single composite metric is interpretable as an overall improvement in mobility. Notably, this allowed us to create a useful visual to demonstrate the changes in subject mobility, which in this study, were positive for all subjects when using the microprocessor knee.

Furthermore, the single metric score can be adopted as a general approach to comprehend multiple scores in individual measures. Applying composite scoring using an autoencoder provides a means to handle data variability and can be beneficial for clinical assessment. However, our dataset was comparatively small, and further investigation is needed with a large dataset utilizing recent deep neural network reduction techniques, such as Improved Complete Ensemble Empirical Mode Decomposition with Adaptive Noise (ICEEDAN) [43,44], Dual Contradistinctive Generative Autoencoder [45], or Stacked LSTM Sequence to Sequence Autoencoder [46]. The proposed composite scoring approach should be explored using different types of datasets.

## 5. Conclusions

This study investigated the approach of using an autoencoder for evaluating two prosthetics. The composite score described the gait quality, and the analysis showed that the condition of the subject was improved when they used the MCK. The results from the learned score are promising and demonstrate the efficacy of the method.

## Figures and Tables

**Figure 1 bioengineering-09-00572-f001:**
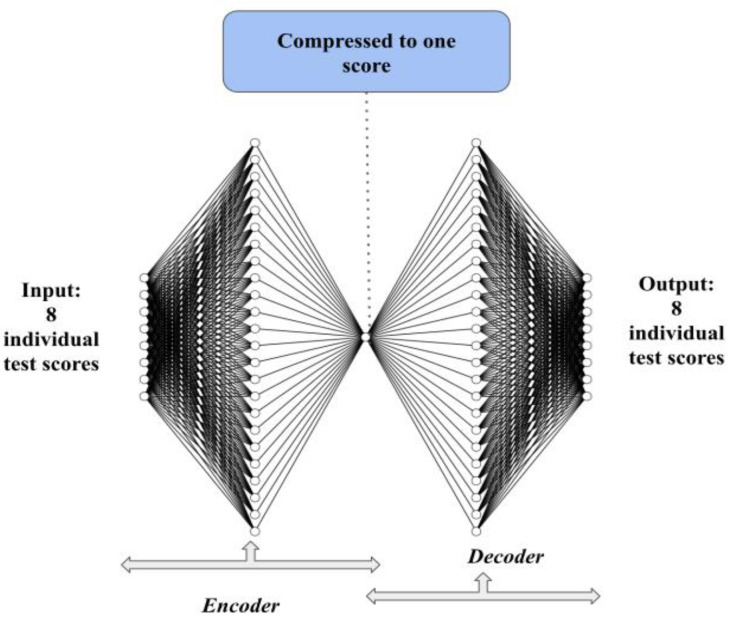
The encoder includes 2 fully connected dense layers and a dropout layer. The first layer is the input layer of 8 test scores. The single central neuron is the compressed score. The decoder contains 2 fully connected dense layers and a dropout layer. The output layer has 8 neurons that retrace the input.

**Figure 2 bioengineering-09-00572-f002:**
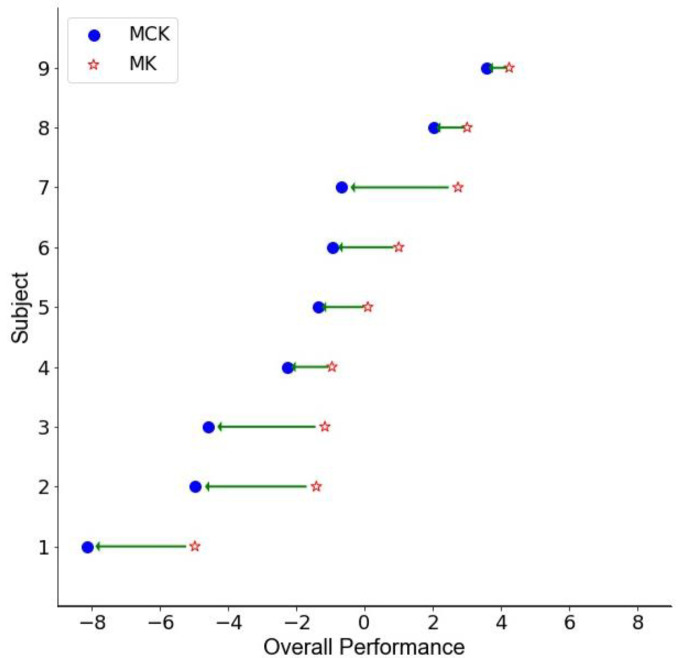
The composite score distills the changes in all 8 measured assessments into one composite score. Note, the decoder accounted for 83.29% of the variation in scores among these subjects. The X-axis corresponds to improvements in all 8 measures for each subject between the MK and MCK conditions, though the order of condition was randomized during the data collection.

**Figure 3 bioengineering-09-00572-f003:**
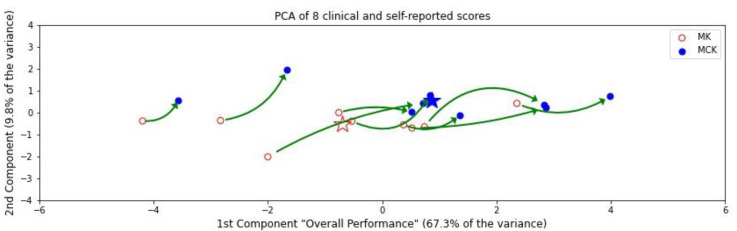
The first principal component, represented by the X-axis, accounts for 67.3% of the variation in scores among these subjects. This axis also corresponds to the improvements in all 8 measures as the subjects move along the X-axis. The Y-axis only accounts for the remaining 9.8% and was only included as an additional dimension for visualization. Stars indicate the mean values for each device.

**Table 1 bioengineering-09-00572-t001:** Autoencoder architecture with hyper-parameters.

Layer	Layer Type	Number Nodes	Activation
1	Input	8	
2	Dense	24	relu
3	Dropout	0.4	
4	Dense	1	linear
5	Dropout	0.4	
6	Dense	24	relu
7	Output	8	

**Table 2 bioengineering-09-00572-t002:** Means and standard deviations (std) of test results for each condition.

Test	Conditions	
	MK (Mean ± Std)	MCK (Mean ± Std)
6 MWT	137.48 ± 85.63	145.43 ± 110.30
10 MWT	22.27 ± 9.72	15.3 ± 7.24
BERG	37.11 ± 7.5	43.56 ± 13.20
FSST	17.37 ± 5.04	16.79 ± 11.17
TUG	27.47 ±14.96	25.32 ± 14.14
AMP	31.44 ± 7.04	35.67 ± 5.40
MFES	7.78 ± 1.14	9.33 ± 0.69
PEQ-amb	60.63 ± 18.75	81.92 ± 18.74
AE	0.29 ± 0.89	−1.92 ± 1.13

**Table 3 bioengineering-09-00572-t003:** Load factors for the first PCA component used as the X-axis in Figure 3.

Test	Load Factors	Direction of Improvement in Scores
AMP	0.395	Higher
BERG	0.388	Higher
6 MWT	0.332	Higher
PEQ-amb	0.313	Higher
MFES	0.286	Higher
FSST	−0.355	Lower
10 MWT	−0.362	Lower
TUG	−0.383	Lower

## Data Availability

De-identified data are available from the authors upon request.

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
