# Peer review of "Autoencoder Composite Scoring to Evaluate Prosthetic Performance in Individuals with Lower Limb Amputation"

_bioengineering, 2022, doi:10.3390/bioengineering9100572_

Round 1
Reviewer 1 Report
This paper proposed a novel approach to predict clinical outcomes based on related features small samples. The deep autoencoder is utilized to reduce the dimensionality and predict the outcome. Overall, this is an interesting research with certain level of technical novelty. However, major revision is needed at current stage for the following issues:
First, the rational of using deep autoencoder is weak. The authors need to discuss why they have to apply deep autoencoder to reduce the dimensionality.
Second, is deep autoencoder considered a linear or nonlinear dimension reduction? Why is it more suitable in comparison with PCA, SVD, or others?
Third, the amount of the experimental results is too few. The depth of this research needs to be enhanced.
The details of training the deep autoencoder is missing. What is the loss function? What is the training strategy? Please be specific
Moreover, how does the authors determine the hyperparameters of deep autoencoder? For example, what is optimal size of the number of hidden layers? What is the activation function?
What is the optimization strategy of the training process?
Another popular decomposition approach is called ICEEMDAN which has been utilized in a lot of applications. Please cite the following two papers and discuss the possibility of using ICEEMDAN in your dataset:
Ghimire, S., Deo, R. C., Casillas-Pérez, D., & Salcedo-Sanz, S. (2022). Improved Complete Ensemble Empirical Mode Decomposition with Adaptive Noise Deep Residual model for short-term multi-step solar radiation prediction. Renewable Energy, 190, 408-424.
Li, H., Deng, J., Feng, P., Pu, C., Arachchige, D. D., & Cheng, Q. (2021). Short-term nacelle orientation forecasting using bilinear transformation and ICEEMDAN framework. Frontiers in Energy Research, 9, 780928.
Reviewer 2 Report
Dear Authors,
Many thanks for your paper submission to MDPI Journal of Bioengineering. This paper had some useful components, however, to be frank, the problems are quite explicit, which must be carefully fixed. I specify the potential issues as below (may not limited to all of these).
1) Abstract: It is too long (over 250 words, close to 300). Please condense this part to 180~200 words in total. The claims on measure and statistical tests should be significantly shortened. The concluding remarks should address the keynote quantitative scores with respect to experimental results.
2) Introduction: It lacks both the paragraph on summary of your major contributions (3-4 manifolds of your work, with a little bit specific details), and the organization on the remainder of this paper. Given the current version only takes up 10 pages, the authors either have to condense the historical review, or later expand the part of approach and proposed tests.
3) Methods: In Subsection 2.4, the architecture of your encoder is too simple (Line 170), the workflow of your proposed approach is not clear. I also think the narrations of the rest subsections are a bit too generic. Please consider applying the required edits. Thanks a lot!
4) The authors missed to present a section on the evaluation metrics along with proposed scheme (which might need some mathematical derivations), the readers might not be clear with your algorithmic model or actual work.
5) Results: The composite scores are displayed in two figures, and table 1 presents the loaded factors for the first PCA components. I think these are not the wise ways to present convincible quantitative analysis. Is there any alternative ways to show your experiments and results? Thanks~
6) Figures and Tables: Limited set of work, and the size of Figure 2 is too large. Visual results make little sense to contribute to the actual topic. The tabulated results are also quite limited. Please consider turning over this section and make decent updates with respect to newly arranged tests.
7) Conclusions: The authors had a solid section on discussion, while the whole conclusion part is missing. Please specify your conclusions and future work, which should be comprised of at least 3 parts: main summary of their work, opening problems to be solved (or summary of research challenges), and brief summary of research orientations on future study. Thanks a lot!
8) References: several aspects of revisions must be performed: a) apply abbreviated style on the title of some journals; b) a few more latest publications in Years 2019-2022 which are similar / parallel to your study area (machine learning and deep learning based schemes on composite scoring of performance evaluation of prosthetics), are recommended to be supplemented; c) Citations on conference proceedings should be loaded with time duration, location, and page numbers. Please follow the template and update with required changes. Thanks a lot!
9) Other minor problematic issues must be addressed in your revisions:
a) The literal quality is barely acceptable, while use of English should be improved in the updated version. A few minor grammatical issues also need to be fixed. Please conduct proofreading on the related context carefully.
b) Align the size of figures (a few of them are a bit too large) and the title of some figures must be more specific. Align the intervals on line of tables.
c) Remove some reduntant half-space and full-space before a comma or a full-stop, i.e., the start of Line 210, and the middle position at Line 23, etc.
d) Some of the characters in terms, must be italic, i.e., p-value.
Once again, thank you for the manuscript submission to MDPI Journal of Bioengineering. We appreciate your future efforts and wish you good luck to improve the overall quality of your research work. All the best!
Stay well,
Yours faithfully,
Round 2
Reviewer 1 Report
The authors have made great improvement over the manuscript and it can be considered for publication for now.